# IMPQ: Interaction-Aware Layerwise Mixed Precision Quantization for LLMs

## Abstract

Large Language Models (LLMs) promise impressive capabilities, yet their multi-billion-parameter scale makes on-device or low-resource deployment prohibitive. Mixed-precision quantization offers a compelling solution, but existing methods struggle when the average precision drops below four bits, as they rely on isolated, layer-specific metrics that overlook critical inter-layer interactions affecting overall performance. In this paper, we propose two innovations to address these limitations. First, we frame the mixed-precision quantization problem as a cooperative game among layers and introduce **S**hapley-based **P**rogressive **Q**uantization **E**stimation (**SPQE**) to efficiently obtain accurate Shapley estimates of layer sensitivities and inter-layer interactions. Second, building upon SPQE, we propose **I**nteraction-aware **M**ixed-**P**recision **Q**uantization (**IMPQ**), which translates these Shapley estimates into a binary quadratic optimization formulation, assigning either 2- or 4-bit precision to layers under strict memory constraints. Comprehensive experiments conducted on Llama-3, Gemma-2, and Qwen-3 models across three independent PTQ backends (Quanto, HQQ, GPTQ) demonstrate IMPQ's scalability and consistently superior performance compared to methods relying solely on isolated metrics. Across average precisions spanning 4 bits down to 2 bits, IMPQ cuts Perplexity by 20 – 80 % relative to the best baseline, with the margin growing as the bit-width tightens.

## 1 Introduction

LLMs have shown impressive performance across various NLP tasks, including text generation, reasoning, and question answering (OpenAI et al., 2024; Touvron et al., 2023). However, their effectiveness is closely tied to increasing model scales, now often reaching hundreds of billions or trillions of parameters (Brown et al., 2020). This massive size creates significant memory and computational demands, limiting deployment on resource-constrained devices such as mobiles, edge sensors, or standard GPUs.

Quantization effectively compresses LLMs to reduce these deployment challenges. Among quantization techniques, Post-Training Quantization (PTQ) is particularly useful, compressing models and accelerating inference without costly retraining (Yao et al., 2024). Early PTQ approaches uniformly applied bit-widths to model weights and activations (Jacob et al., 2017). Techniques like SmoothQuant improved uniform quantization by smoothing activation outliers (Xiao et al., 2024), yet uniform quantization does not fully exploit layer-specific precision requirements in LLMs. Mixed-precision PTQ addresses layer heterogeneity by assigning different bit-widths across model layers. For example, critical layer weights might remain at 4 bits, while less sensitive layers use 2 bits, substantially reducing model size without retraining (Dettmers et al., 2023; Frantar et al., 2023; Lin et al., 2024). Existing mixed-precision schemes typically determine bit allocation using isolated metrics such as weight distributions, cosine similarity, activation sensitivity, or layer-specific scores (Dumitru et al., 2024b; Li et al., 2023a; Sarhaddi et al., 2025; Hu et al., 2024). Some approaches consider second-order information such as Hessians (Dong et al., 2019), but calculating Hessians for large LLMs remains computationally challenging. While beneficial, these methods often overlook how quantization errors propagate through the network, potentially misallocating high-precision resources and impairing overall effectiveness.

To overcome limitations associated with conventional layer-wise heuristics in mixed-precision quantization, we frame this problem as a cooperative game among LLM layers and leverage Shapley value analysis (Shapley, 1953; Ghorbani & Zou, 2020) to evaluate each layer's expected marginal contribution under quantization-induced interactions. Specifically, we define the game's payoff as the change in per-token negative log-likelihood resulting from quantization. Direct computation of Shapley values is computationally prohibitive for LLMs; hence, we employ Monte-Carlo permutation sampling for efficient approximation.

Unlike prior interpretability approaches that measure layer contributions by complete pruning—an approach known to severely degrade performance and result in unreliable, high-variance Shapley estimates (Zhang et al., 2024)—we propose Shapley-based Progressive Quantization Estimation (SPQE). SPQE uniformly quantizes the model to a moderate baseline precision and then progressively reduces the precision of each layer to a lower precision within each Monte-Carlo sampled permutation. This progressive strategy maintains model stability, allowing incremental rather than catastrophic performance degradation. Consequently, our approach yields accurate and low-variance Shapley estimates.

Building upon these Shapley estimates from SPQE, we introduce **I**nteraction-aware **M**ixed-**P**recision **Q**uantization (IMPQ), a novel framework for optimal precision assignment. IMPQ converts inferred layer sensitivities and inter-layer interactions into a quadratic surrogate model that predicts the loss increase resulting from assigning either 2-bit or 4-bit precision to individual layers. Minimizing this surrogate under predefined memory constraints yields a binary quadratic optimization problem, where each binary variable determines the bit-width assigned to a specific layer. To efficiently solve this problem, we linearize the quadratic objective into a Mixed-Integer Linear Program (MILP), enabling standard optimization solvers to obtain globally optimal bit assignments.

Our contributions are:

- We propose SPQE, an efficient method leveraging cooperative game theory and progressive quantization to accurately estimate layer sensitivities and inter-layer interactions in mixed-precision quantization.
- We introduce IMPQ, a novel optimization framework that translates these layer sensitivity estimates into optimal bit-width assignments via MILP.
- We conduct comprehensive ablation analyses examining how the number of permutations sampled in SPQE and the inclusion of inter-layer interaction terms impact quantization performance, providing critical insights for practical implementation.

Extensive evaluations on widely adopted models—including Llama-3, Gemma-2, and Qwen-3—across three independent PTQ frameworks (Quanto, HQQ, GPTQ) demonstrate that IMPQ consistently achieves superior performance compared to conventional methods relying on isolated, layer-specific metrics for mixed-precision quantization at equivalent memory budgets.

## 2 RELATED WORKS

### 2.1 MIXED-PRECISION QUANTIZATION AND LAYER SENSITIVITY.

Quantization methods for LLMs aim to reduce computational and memory overhead by representing parameters at lower precision, typically ranging from 2 to 8 bits Choi et al. (2018); Hubara et al. (2021); Yao et al. (2022); Park et al. (2022); Gholami et al. (2022); Xi et al. (2023). Post-Training Quantization (PTQ) is particularly appealing due to its efficiency, as it quantizes pre-trained models without requiring retraining. PTQ techniques include static quantization, which uses calibration datasets, and dynamic quantization, where scales are computed on-the-fly during inference Banner et al. (2019); Zhu et al. (2023).

Recent research explores mixed-precision quantization strategies, assigning varying bit-widths across layers based on their sensitivity. For instance, LLM-MQ Li et al. (2023b) employs gradient-based sensitivity analysis, while TinyAgent Chen et al. (2024a) integrates TrimLLM (Hu et al., 2024) and AWQ (Lin et al., 2024) with selective layer freezing to maintain accuracy. Methods like ResQ Saxena et al. (2024) and CMPQ Chen et al. (2024b) enhance mixed-precision quantization using low-rank residuals and channel-wise statistics, improving overall performance and

hardware efficiency. Additionally, HAWQ Dong et al. (2019) leverages Hessian-based sensitivity analysis, surpassing simpler sensitivity metrics. Dumitru et al. Dumitru et al. (2024b) propose meta-layerwise quantization strategies, employing explicit metrics such as Layer Input Modification and Z-score Distribution to allocate bit-width flexibly under memory constraints, effectively complementing techniques like GPTQ Frantar et al. (2023) and Quanto Quanto (2024).

### 2.2 SHAPLEY-BASED LAYER IMPORTANCE

Existing quantization methods typically assess layers sensitivities independently using heuristics like norm-based metrics or Hessian approximations, neglecting inter-layer dependencies. Recent research integrates cooperative game theory, specifically Shapley values Shapley (1953), to quantify layer importance based on marginal contributions across various subsets. For example, Neuron Shapley (Ghorbani & Zou, 2020) uses Monte Carlo sampling to estimate how individual neurons contribute to a network's performance, and finds that removing neurons with the highest Shapley values severely degrades accuracy.

For LLMs, previous research has effectively applied Shapley value analysis to identify critical layers influencing model Perplexity (Zhang et al., 2024). These studies primarily utilized Shapley values for structured pruning, demonstrating improved pruning efficacy and model interpretability compared to simpler heuristic methods (Sun et al., 2025; Adamczewski et al., 2024a;b). However, these approaches rely heavily on layer pruning strategies, significantly limiting their application to post-training quantization. Specifically, pruning leads to rapid performance degradation, causing high variance in Shapley value estimates and restricting the number of layers that can be effectively analyzed for interactions.

In contrast, our approach, SPQE, addresses this limitation by introducing the first practical application of Shapley value analysis tailored specifically for post-training mixed-precision quantization. By replacing abrupt layer pruning with progressive quantization, we ensure gradual performance changes, resulting in lower variance Shapley estimates and allowing for more extensive consideration of inter-layer interactions.

## 3 METHODS

### 3.1 THE SHAPLEY-BASED PROGRESSIVE QUANTIZATION ESTIMATION (SPQE)

In this work, we propose the Shapley-based Progressive Quantization Estimation (SPQE), a progressive quantization scheme designed within a Shapley value framework to evaluate Transformer layer importance for LLMs. Traditional pruning methods and direct quantization from full precision typically degrade model performance significantly and introduce high variance in Shapley estimates. In contrast, SPQE maintains model stability, enabling accurate and low-variance Shapley value assessments. Each Transformer layer acts as a "player" in a cooperative game, where quantization from high to low precision represents the explicit "removal" of a player.

Shapley values, grounded in cooperative game theory, provide a principled way to quantify each player's contribution to a team effort by averaging their marginal contributions across all possible coalitions (Shapley, 1953). Formally, for a set of $n$ players with value function $v(\cdot)$, the Shapley value $\phi_i$ for player $i$ is defined as the average payoff difference when $i$ joins a coalition $S$ that does not include $i$:

$$\phi_i = \sum_{S \subseteq N \setminus \{i\}} \frac{|S|!(n - |S| - 1)!}{n!} \Big( v(S \cup \{i\}) - v(S) \Big) \tag{1}$$

we represent an LLM as an ordered set $T = \{1, 2, \ldots, L\}$ of layers. For a subset $S \subseteq T$ of layers, we define $S$ as a set of layers with high precision while others are quantized to low precision. Specifically, for a layer $t \in T$, its precision $b_t$ is determined by:

$$b_t = \begin{cases} b_{\text{high}} & \text{if } t \in S \\ b_{\text{low}} & \text{if } t \in T \setminus S \end{cases} \tag{2}$$

where $b_{\text{high}}$ and $b_{\text{low}}$ represent the high and low bit precisions (4- and 2-bit respectively). This formulation allows us to systematically evaluate how different layer combinations affect model performance under quantization.

We use the average per-token negative log-likelihood (NLL) as pay-offs when estimating Shapley values.

$$v_{\text{NLL}}(S) = \mathbb{E}_{(x,t)\sim\mathcal{D}}\big[-\log p\big(x_{t+1} \mid x_{\leq t}; S\big)\big] \tag{3}$$

where $\mathcal{D}$ is the validation corpus.

To efficiently estimate Shapley values, we adopt Monte-Carlo permutation sampling. Specifically, we sample $M$ random permutations of the layers. For each permutation $\pi = (\pi_1, \pi_2, \ldots, \pi_L)$, which represents a random ordering of the layer indices $\{1, 2, \ldots, L\}$, we start with uniformly quantizing all layers in $b_{\text{high}}$ and progressively quantize a layer to $b_{\text{low}}$ according to the permutation order. At each quantization step of quantizing layer $\pi_\ell$, we denote the set of layers that remain at $b_{\text{high}}$ by:

$$S_{\ell+1} = \{\pi_{\ell+1}, \ldots, \pi_L\} \tag{4}$$

When quantizing layer $\ell$ from $b_{\text{high}}$ to $b_{\text{low}}$, its marginal contribution to the model's value function is explicitly computed as the immediate change in the value function due to reducing this specific layer's precision:

$$\Delta v_\ell = v(S_\ell) - v(S_\ell \setminus \{\pi_\ell\}) = v(S_\ell) - v(S_{\ell+1}) \tag{5}$$

After performing this calculation for all permutations and positions, we approximate the Shapley value for layer $i$, denoted $\hat{\phi}_i$, by averaging its marginal contributions across the $M$ permutations:

$$\hat{\phi}_i = \frac{1}{M} \sum_{m=1}^{M} \Delta v_i^{(m)} \tag{6}$$

This method effectively captures both individual layer sensitivity and inter-layer interactions under progressive quantization.

### 3.2 Interaction-Aware Mixed Precision Quantization (IMPQ)

Building upon SPQE, we now propose an extended approach explicitly designed for interaction-aware mixed-precision quantization. Our goal is to optimally assign each Transformer layer either 2-bit or 4-bit precision by explicitly accounting for both individual layer sensitivities and cross-layer interactions.

#### Second-Order Taylor Analysis

Consider a Transformer with layers indexed by the ordered set $T = \{1, 2, \ldots, L\}$. Quantizing layer $i$ introduces a perturbation $\epsilon_i$ to its weights, yielding perturbed weights $\widetilde{\mathbf{W}}_i = \mathbf{W}_i + \epsilon_i$. The resulting change in loss $\Delta L$ admits the second-order Taylor approximation

$$\Delta L \approx \sum_{i=1}^{L} \mathbf{g}_i^{\top} \epsilon_i \;+\; \sum_{i=1}^{L}\sum_{j=1}^{L} \epsilon_i^{\top} \mathbf{H}_{ij}\, \epsilon_j \tag{7}$$

where $\mathbf{g}_i = \nabla_{\mathbf{W}_i} L$ captures linear sensitivity and $\mathbf{H}_{ij} = \nabla^2_{\mathbf{W}_i, \mathbf{W}_j} L$ captures pairwise interactions. Empirically, both terms affect quantization-induced loss, underscoring the need to estimate them explicitly.

#### From Taylor Expansion to Shapley-Based Approximation

Direct evaluation of all $\mathbf{g}_i$ and $\mathbf{H}_{ij}$ is computationally infeasible for LLMs layers. Instead, we leverage SPQE, which empirically estimates the *marginal* loss incurred when each layer is quantized across $M$ random permutations, producing empirical Shapley values $\hat{\phi}_i$.

Specifically, we construct the covariance matrix $\mathbf{C} \in \mathbb{R}^{M \times L}$ from empirical Shapley value deviations, serving as a practical proxy for the Hessian interactions:

$$\mathbf{C} = \frac{1}{M} \left( \Delta v_\ell - \hat{\boldsymbol{\phi}} \right)^\top \left( \Delta v_\ell - \hat{\boldsymbol{\phi}} \right), \quad \hat{\boldsymbol{\phi}} = [\hat{\phi}_1, \ldots, \hat{\phi}_L] \tag{8}$$

Because finite sampling causes high variance in the off-diagonal terms of $\mathbf{C}$, we therefore apply diagonal shrinkage controlled by a hyperparameter $\alpha \in [0, 1]$:

$$\mathbf{K} = (1 - \alpha)\, \mathbf{C} + \alpha \, \mathrm{diag}(\mathbf{C}) \tag{9}$$

where larger $\alpha$ suppresses noisy cross-layer interactions while smaller $\alpha$ preserves them.

Subsequently, we isolate individual first-order sensitivities $\mathbf{a}_i$ by subtracting interaction contributions from empirical Shapley values:

$$\mathbf{a}_i = \hat{\phi}_i - \sum_{j \neq i} K_{ij} \tag{10}$$

### MIXED-INTEGER LINEAR PROGRAMMING FOR BIT ALLOCATION

Given the stabilized layer sensitivities $\mathbf{a}$ and interaction matrix $\mathbf{K}$, we formulate the bit allocation as a constrained quadratic optimization problem.

For each layer $i$, we introduce a binary decision variable $q_i \in \{0, 1\}$, where $q_i = 1$ indicates the layer remains at low precision and $q_i = 0$ means it is promoted to high precision. Our objective is to minimize the approximated total loss increase induced by quantization, expressed through a quadratic function involving both linear sensitivities and pairwise interactions:

$$\Delta L(\mathbf{q}) = \mathbf{a}^\top \mathbf{q} + \mathbf{q}^\top \mathbf{K} \mathbf{q} \tag{11}$$

where $\mathbf{q} = (q_1, \cdots, q_L)$.

To respect the memory constraints $\mathbf{B}$ given the byte cost $c_i$ for each layer to be promoted from lower-bit to higher-bit, we impose a linear constraint limiting the number of layers maintained at high precision. Putting these goals together, the resulting optimization is a binary quadratic programming problem:

$$\min_{\mathbf{q} \in \{0,1\}^L} \Delta L(\mathbf{q}) \quad \text{s.t.} \quad \sum_{i=1}^{L} c_i (1 - q_i) \leq \mathbf{B} \tag{12}$$

We solve this quadratic optimization by reformulating it into an equivalent Mixed-Integer Linear Program. To linearize the quadratic term $q_i q_j$, we introduce auxiliary binary variables $y_{ij}$ representing pairwise interactions, enforcing linear constraints:

$$y_{ij} \geq q_i + q_j - 1, \quad y_{ij} \leq q_i, \quad y_{ij} \leq q_j, \quad y_{ij} \in \{0, 1\} \tag{13}$$

ensuring $y_{ij} = 1$ if and only if $q_i = q_j = 1$. This standard linearization transforms the quadratic objective into a linear one in terms of $q$ and auxiliary variables $y$, enabling efficient solution via standard MILP solvers.

## 4 EXPERIMENTS

We evaluate IMPQ on three model families: Gemma-2 (2B, 9B) Team et al. (2024), Llama-3 (3.2B, 8B) Grattafiori et al. (2024), and Qwen3 (4B, 8B) Yang et al. (2025). Our evaluation focuses on layerwise mixed-precision quantization, where we constrain the target model's average bit-width to a range between 2 and 4 bits. The diagonal shrinkage hyperparameter $\alpha$ is set to 0.5 across all experiments.

To benchmark performance, we compare our method against three PTQ baselines: Quanto Quanto (2024), HQQ Badri & Shaji (2023), and GPTQ Frantar et al. (2022). For Quanto and HQQ, we apply a uniform scaling factor. This simple, calibration-free scaling allows for rapid quantization, though it may result in worse quantization performance compared to the more time-consuming and resource-intensive GPTQ method. We choose Quanto for our SPQE across all the models in our experiments because it's efficient in-place weight quantization and rapid layer processing are critical for the efficiency of our estimation approach. Finally, we use SCIP Bolusani et al. (2024) as our MILP solver. All experiments were conducted on a server with two NVIDIA A40 GPUs using a fixed seed for reproducibility.

## 4.1 EVALUATION CONFIGURATIONS

To evaluate our layer-wise quantization performance, we use Perplexity as our major evaluation metric. Our evaluation framework uses different datasets for distinct purposes: Shapley value estimation, and final performance assessment.

For Shapley value estimation purposes, we use the C4 dataset Raffel et al. (2020). We use the training split of C4 for SPQE calibration and final bit allocation optimization, while the WikiText-2 validation split Merity et al. (2016) is used for the final quantization evaluation, providing unbiased comparisons of language modeling performance across different quantization strategies.

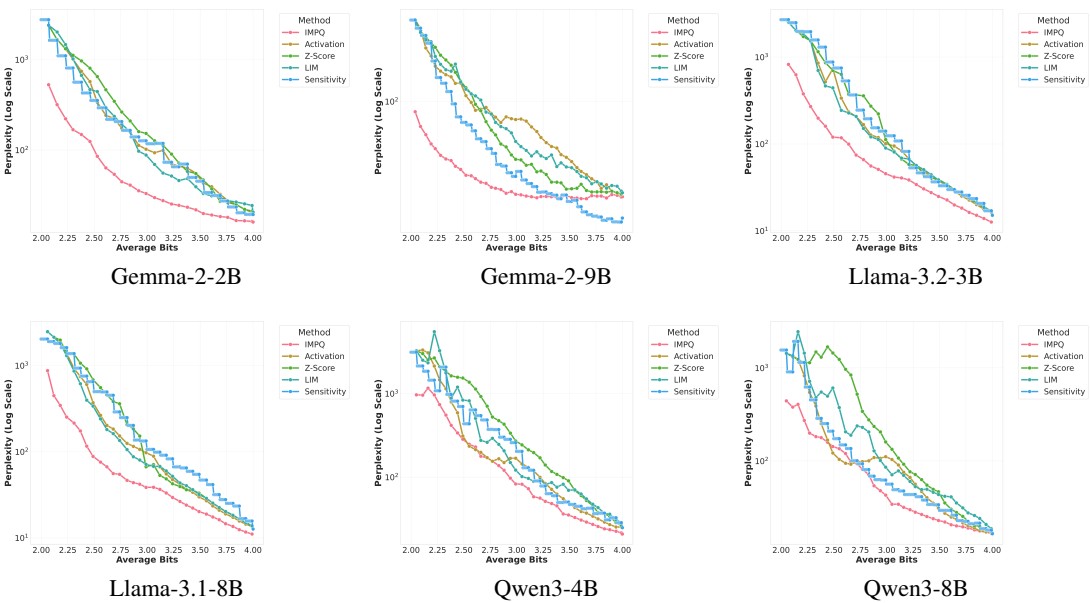

Figure 1: Wikitext-2 Perplexity comparison of quantization methods across Gemma, Llama, Qwen models on GPTQ.

We compare IMPQ against the following layer-wise mixed precision quantization methods:

- **LLM-MQ Sensitivity** Li et al. (2023a): Uses first-order Taylor approximations to measure per-layer quantization sensitivity; assigns bit-widths to minimize performance loss.
- **LIM (Layer Input Modification)** Dumitru et al. (2024a): Scores layer importance via the negative cosine similarity between input and output embeddings on a calibration set; larger change suggests higher importance.
- **ZD (Z-score Distribution)** Dumitru et al. (2024a): Assesses importance by the proportion of outlier weights in the target layer; no calibration data required. More outliers suggest greater importance.
- **Activation-based Scoring** Kong et al.: Uses the Frobenius norm of layer activations; larger norms suggest more critical layers.

## 4.2 RESULT ANALYSIS

Figure 1 and Table 1 illustrate comprehensive Perplexity comparisons across quantization methods including Activation, Sensitivity, LIM, Z-Score, and IMPQ for multiple LLMs. The consistently superior performance of IMPQ can be attributed to its effective integration of layer interaction effects into the quantization process. Unlike baseline methods, which primarily assess layers in isolation, IMPQ explicitly accounts for how quantization errors propagate through the network, thus significantly reducing the overall Perplexity.

**Performance across Quantization bit-widths.** Across the models tested under GPTQ quantization, IMPQ consistently delivers the lowest Perplexity values, with its advantages becoming more

pronounced as bit precision decreases. In the lowest range of 2.01–2.5 bits, IMPQ achieves a Perplexity of 233.98 in Gemma-2B, representing a reduction of more than 79% and 81% relative to Sensitivity at $1.12 \times 10^3$ and LIM at $1.25 \times 10^3$, respectively. Similar improvements are seen in Gemma-2-9B, where IMPQ achieves 48.52 compared to Sensitivity's 189.55 and LIM's 214.03—reductions of approximately 74% and 77%. As illustrated in Figure 1, this trend holds consistently across the entire curve for Gemma-2-9B, where IMPQ maintains the lowest Perplexity across nearly all bit-width, especially under more severe quantization constraints. Similarly, in Qwen3-4B, IMPQ maintains a Perplexity of 697.28, substantially outperforming Sensitivity at $1.56 \times 10^3$ and LIM at $2.38 \times 10^3$ by margins of 55% and 71%. This further highlights IMPQ's robustness under aggressive quantization and its ability to scale effectively across architectures of varying scale and complexity.

As the bit budget increases, IMPQ continues to retain favorable Perplexity values with GPTQ quantization. For example, at 2.5–3.0 bits, Llama-3.2-3B achieves a score of 73.11 with IMPQ, a 79% reduction relative to Sensitivity's 343.64. Similarly, Qwen3-8B records 82.74 with IMPQ compared to Activation's 101.55, reflecting a 19% improvement. Even in the highest precision range, 3.5–3.99 bits, IMPQ remains competitive. Llama-3.2-3B sees a Perplexity of 17.08, outperforming Z-Score's 24.84 by 31%, and Qwen3-8B's 19.03, slightly better than Activation's 21.06. These results show that IMPQ scales across diverse architectures and maintains low Perplexity under tighter bit constraints, as shown in Figure 1, by modeling inter-layer interactions effectively.

| Model | Bit Range | GPTQ | | | | | Quanto | | | | | HQQ | | | | |
|---|---|---|---|---|---|---|---|---|---|---|---|---|---|---|---|---|
| | | Act. | Sens. | LIM | ZD | IMPQ | Act. | Sens. | LIM | ZD | IMPQ | Act. | Sens. | LIM | ZD | IMPQ |
| Gemma-2-2B | 2.01–2.5 | $1.19 \times 10^3$ | $1.12 \times 10^3$ | $1.25 \times 10^3$ | $1.30 \times 10^3$ | **233.98** | $73.04 \times 10^3$ | $98.70 \times 10^3$ | $90.71 \times 10^3$ | $67.63 \times 10^3$ | **$18.35 \times 10^3$** | $16.52 \times 10^3$ | $21.54 \times 10^3$ | $18.50 \times 10^3$ | $16.17 \times 10^3$ | **$5.85 \times 10^3$** |
| | 2.5–3.0 | 181.40 | 203.67 | 198.16 | 295.38 | **48.35** | $3.74 \times 10^3$ | $1.53 \times 10^3$ | $4.65 \times 10^3$ | $1.78 \times 10^3$ | **397.26** | $1.62 \times 10^3$ | $1.25 \times 10^3$ | $1.61 \times 10^3$ | $1.29 \times 10^3$ | **421.95** |
| | 3.0–3.5 | 72.35 | 79.81 | 50.35 | 83.01 | **24.99** | 218.85 | 247.83 | 110.99 | 253.30 | **54.18** | 161.15 | 231.70 | 108.30 | 236.02 | **49.69** |
| | 3.5–3.99 | 29.19 | 27.63 | 28.00 | 28.20 | **17.70** | 31.53 | 46.55 | 30.87 | 46.54 | **22.09** | 31.57 | 41.09 | 30.77 | 37.45 | **21.96** |
| Gemma-2-9B | 2.01–2.5 | 195.81 | 189.55 | 214.03 | 223.94 | **48.52** | $1.07 \times 10^3$ | $1.07 \times 10^3$ | $1.44 \times 10^3$ | $1.05 \times 10^3$ | **229.67** | 742.81 | 820.53 | $1.05 \times 10^3$ | 797.39 | **250.40** |
| | 2.5–3.0 | 84.15 | 47.82 | 82.68 | 70.12 | **26.26** | 266.84 | 93.51 | 314.96 | 155.49 | **33.68** | 174.87 | 73.31 | 204.10 | 120.87 | **35.47** |
| | 3.0–3.5 | 55.84 | 24.96 | 41.20 | 31.39 | **22.05** | 110.73 | 28.62 | 49.31 | 39.47 | **19.63** | 86.98 | 23.57 | 47.84 | 35.79 | **20.30** |
| | 3.5–3.99 | 28.49 | **16.79** | 28.11 | 24.18 | 21.96 | 32.56 | 16.68 | 26.33 | 25.48 | **19.44** | 30.72 | **15.92** | 26.88 | 25.60 | 20.96 |
| Llama-3.2-3B | 2.01–2.5 | $1.41 \times 10^3$ | $1.81 \times 10^3$ | $1.34 \times 10^3$ | $1.48 \times 10^3$ | **362.59** | $24.05 \times 10^3$ | $13.15 \times 10^3$ | $30.98 \times 10^3$ | $20.00 \times 10^3$ | **$10.20 \times 10^3$** | $10.89 \times 10^3$ | $5.92 \times 10^3$ | $11.44 \times 10^3$ | $8.76 \times 10^3$ | **$5.02 \times 10^3$** |
| | 2.5–3.0 | 185.60 | 343.64 | 164.43 | 334.56 | **73.11** | 643.87 | 791.20 | 729.46 | $1.12 \times 10^3$ | **378.42** | 360.72 | 493.52 | 370.54 | 643.17 | **133.79** |
| | 3.0–3.5 | 61.44 | 70.75 | 58.08 | 55.64 | **33.90** | 67.24 | 79.57 | 63.59 | 72.84 | **38.86** | 51.11 | 59.38 | 45.34 | 50.02 | **31.31** |
| | 3.5–3.99 | 23.63 | 25.56 | 24.61 | 24.84 | **17.08** | 23.99 | 25.99 | 23.67 | 24.04 | **17.43** | 22.12 | 23.25 | 21.48 | 21.74 | **16.05** |
| Llama-3.1-8B | 2.01–2.5 | $1.29 \times 10^3$ | $1.36 \times 10^3$ | $1.21 \times 10^3$ | $1.48 \times 10^3$ | **306.96** | $97.15 \times 10^3$ | $91.01 \times 10^3$ | $100.61 \times 10^3$ | $164.96 \times 10^3$ | **$74.32 \times 10^3$** | $115.95 \times 10^3$ | $47.18 \times 10^3$ | $94.81 \times 10^3$ | $189.80 \times 10^3$ | **$49.70 \times 10^3$** |
| | 2.5–3.0 | 156.09 | 305.34 | 133.11 | 294.18 | **53.02** | 749.91 | $1.64 \times 10^3$ | 522.83 | $74.30 \times 10^3$ | **125.90** | 194.34 | 275.24 | 158.12 | $52.97 \times 10^3$ | **108.28** |
| | 3.0–3.5 | 50.13 | 77.76 | 49.66 | 44.97 | **28.80** | 41.49 | 65.89 | 35.65 | 43.20 | **24.38** | 34.96 | 44.24 | 32.65 | 47.60 | **22.85** |
| | 3.5–3.99 | 19.05 | 28.93 | 20.21 | 20.27 | **14.57** | 17.81 | 28.76 | 17.39 | 17.57 | **14.03** | 17.57 | 21.26 | 17.17 | 17.09 | **13.41** |
| Qwen3-4B | 2.01–2.5 | $1.75 \times 10^3$ | $1.56 \times 10^3$ | $2.38 \times 10^3$ | $2.22 \times 10^3$ | **697.28** | $412.37 \times 10^3$ | $548.73 \times 10^3$ | $257.78 \times 10^3$ | $928.68 \times 10^3$ | **$127.86 \times 10^3$** | $105.06 \times 10^3$ | $116.57 \times 10^3$ | $103.93 \times 10^3$ | $182.17 \times 10^3$ | **$53.58 \times 10^3$** |
| | 2.5–3.0 | 180.44 | 408.11 | 322.61 | 687.04 | **157.65** | $116.54 \times 10^3$ | $42.63 \times 10^3$ | $20.72 \times 10^3$ | $1.21 \times 10^6$ | **$4.96 \times 10^3$** | $18.40 \times 10^3$ | $30.55 \times 10^3$ | $24.17 \times 10^3$ | $44.38 \times 10^3$ | **$10.82 \times 10^3$** |
| | 3.0–3.5 | 90.69 | 94.18 | 86.38 | 152.59 | **54.01** | $2.07 \times 10^3$ | $2.14 \times 10^3$ | $1.05 \times 10^3$ | $14.42 \times 10^3$ | **417.44** | 313.47 | $9.28 \times 10^3$ | 401.89 | $3.56 \times 10^3$ | **134.40** |
| | 3.5–3.99 | 32.52 | 38.26 | 46.05 | 43.82 | **26.42** | 122.50 | 240.26 | 97.68 | 577.45 | **49.10** | 51.39 | $1.74 \times 10^3$ | 57.13 | 254.76 | **32.07** |
| Qwen3-8B | 2.01–2.5 | 689.03 | 794.99 | $1.04 \times 10^3$ | $1.36 \times 10^3$ | **258.60** | $808.51 \times 10^3$ | $912.62 \times 10^3$ | $185.09 \times 10^3$ | $149.74 \times 10^3$ | **$73.64 \times 10^3$** | $285.77 \times 10^3$ | $519.82 \times 10^3$ | $330.15 \times 10^3$ | $240.00 \times 10^3$ | **$324.80 \times 10^3$** |
| | 2.5–3.0 | 101.55 | 103.07 | 195.68 | 533.85 | **82.74** | $102.53 \times 10^3$ | $422.60 \times 10^3$ | $7.57 \times 10^3$ | $43.39 \times 10^3$ | **$1.05 \times 10^3$** | $53.86 \times 10^3$ | $247.68 \times 10^3$ | $16.04 \times 10^3$ | $25.98 \times 10^3$ | **$1.31 \times 10^3$** |
| | 3.0–3.5 | 60.77 | 42.92 | 57.82 | 77.25 | **28.53** | 623.05 | $25.50 \times 10^3$ | 518.57 | $2.14 \times 10^3$ | **107.02** | 174.20 | $19.54 \times 10^3$ | 259.85 | $1.54 \times 10^3$ | **70.21** |
| | 3.5–3.99 | 21.06 | 23.37 | 29.88 | 24.27 | **19.03** | 51.51 | 246.10 | 38.07 | 145.58 | **26.82** | 30.41 | 450.06 | 29.59 | 89.52 | **22.32** |

Table 1: Wikitext-2 Perplexity comparison across GPTQ, Quanto, and HQQ quantization baselines.

**Performance across Quantization Methods.** In addition to GPTQ, IMPQ consistently outperforms alternative baselines under both HQQ and Quanto quantization techniques. For instance, when applying HQQ quantization to the Llama-3.1-8B model, IMPQ achieves an average Perplexity improvement of 32.7% compared to the strongest baseline methods across all evaluated bit-widths. Similarly, for the Gemma-2-9B model quantized using Quanto, IMPQ yields a substantial average Perplexity reduction of 52.3% relative to the best-performing baselines across all considered bit-width.

Notably, these gains become even more pronounced at lower bit-width, consistent with the trend observed under GPTQ. Specifically, when employing Quanto quantization on Gemma-2-9B at the most constrained bit range (2.01–2.5 bits), IMPQ significantly reduces Perplexity by an average of 78.5% compared to the best baseline method (Sensitivity). Likewise, for the Llama-3.1-8B model under HQQ quantization at similarly low bit-width (2.01–2.5 bits), IMPQ attains a 47.6% Perplexity reduction over the LIM baseline. These results emphasize IMPQ's robustness and effectiveness in preserving model performance, particularly under aggressive quantization conditions.

However, the Sensitivity baseline underperforms on Llama and Qwen models under Quanto and HQQ quantization due to its early selection of layers highly sensitive to quantization. This issue arises from the layer-selection strategy rather than from the quantization methods themselves, as confirmed by the better performance of other baselines.

Overall, these results clearly indicate IMPQ's superior quantization performance. By explicitly capturing these interactions, IMPQ efficiently mitigates accumulated quantization errors, yielding significantly lower perplexities across various models and quantization precisions.

# 5 Ablation Study

We conduct a comprehensive ablation study to analyze the impact of key hyperparameters in our SPQE framework, specifically examining how the number of Monte Carlo sampling affects quantization performance and layer importance estimation accuracy.

| Sampling | Avg PPL | Rel. $\Delta$ Avg (%) | Rel. $\Delta$ Geo. Mean (%) |
|---|---|---|---|
| 10 | $19.78 \times 10^3$ | NaN | NaN |
| 20 | $19.77 \times 10^3$ | -0.04% | -4.10% |
| 30 | $19.49 \times 10^3$ | -1.46% | -12.66% |
| 40 | $19.27 \times 10^3$ | -2.58% | -16.17% |
| 50 | $19.22 \times 10^3$ | -2.84% | -19.93% |
| 60 | $19.26 \times 10^3$ | -2.60% | -18.61% |
| 70 | $19.37 \times 10^3$ | -2.04% | -15.06% |
| 80 | $19.28 \times 10^3$ | -2.55% | -17.88% |
| 90 | $19.27 \times 10^3$ | -2.59% | -18.83% |
| 100 | $19.26 \times 10^3$ | -2.65% | -19.88% |

Table 2: WikiText-2 Perplexity Analysis vs SPQE Sampling on Quanto

**Effect of SPQE Sampling.** A critical hyperparameter in SPQE is the number of Monte Carlo permutation samples used to estimate Shapley values. Unlike prior Shapley-based layer importance approaches that rely on ablating entire layers–which often induces catastrophic performance degradation and noisy estimates–our SPQE method quantizes layers progressively, resulting in much smoother performance changes. This gradual degradation enables lower-variance Shapley estimates, allowing meaningful signals even with relatively few samples.

We evaluate the impact of SPQE sample count on quantization quality using LLaMA 3.1-8B across a range of 10 to 100 SPQE samples. Table 2 illustrates how increasing the number of Monte Carlo samples affects the quantized model's Perplexity on the WikiText-2 validation set using Quanto quantization. As the sample count grows, the model's post-quantization Perplexity improves steadily, reflecting more precise Shapley value estimates that better capture layer sensitivities and inter-layer interactions.

The **relative delta** measures the percentage change in a metric relative to the baseline (10 samples). The **geometric mean relative delta** summarizes a distribution of Perplexity values using the geometric mean, which is effective for data spanning several orders of magnitude. This metric quantifies the overall change in model performance against the baseline, indicating both the magnitude and consistency of improvement.

Notably, even with as few as 10 random SPQE samples, clear layer importance patterns emerge. For instance, the first and final transformer layers consistently appear as highly sensitive to quantization across different models. This demonstrates that SPQE can capture fundamental layer importance signals with minimal computational overhead. However, the returns diminish at higher sample counts: beyond roughly 50 samples, additional samples yield diminishing improvements. The relative delta for average Perplexity shows a maximum improvement of -2.84% at 50 samples, with only marginal further gains to -2.65% at 100 samples. Similarly, the geometric mean relative delta reaches its maximum improvement of -19.93% at 50 samples, with only marginal further gains to -19.88% at 100 samples. After 90 samples, the changes become negligible: the relative delta changes by only 0.06% (from -2.59% to -2.65%) and the geometric mean changes by only 0.05% (from -18.83% to -19.88%). This convergence behavior provides a clear stopping criterion, indicating that both metrics have effectively converged. Consequently, we adopt 100 samples in all main experiments as a practical sweet spot, achieving near-maximal Perplexity improvement while keeping the computational overhead manageable.

**SPQE vs. Layer Pruning.** To further show the advantages of SPQE over conventional layer pruning for Shapley value estimation, we conduct a comparative analysis using the Llama 3.1-8B model. As illustrated in Figure 2, the pruning-based approach results in a rapid and uncontrolled escalation of Perplexity, reaching near-infinite values after the removal of only a few layers. This phenomenon renders the marginal contribution estimates highly unstable and uninformative, thereby impeding the reliable computation of both individual layer importance and inter-layer interactions within the Shapley framework. The resulting high variance in Shapley estimates ultimately degrades the quality of mixed-precision bit allocation, leading to suboptimal quantization performance.

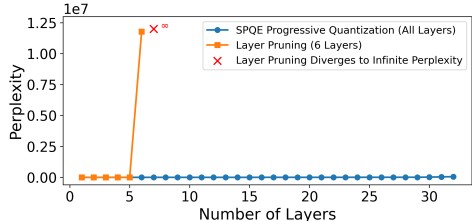

Figure 2: Comparison of perplexity for SPQE and layer pruning-based Shapley estimation on Llama 3.1-8B using Quanto. Layer pruning causes perplexity to diverge after 5 layers, while progressive quantization remains stable.

In contrast, SPQE maintains model stability throughout the quantization process, exhibiting a smooth and gradual increase in Perplexity as layers are progressively quantized from 4-bit to 2-bit precision. This controlled degradation enables the accurate estimation of Shapley values with substantially reduced variance, facilitating robust modeling of both individual and interaction effects across all layers. Empirically, the variance of Shapley estimates under SPQE is significantly lower than that observed with pruning-based methods, supporting more effective and reliable bit allocation in mixed-precision quantization.

## 6 DISCUSSION

Our findings present compelling evidence that the prevailing approach of using isolated, layer-specific metrics is insufficient for effective low-bit quantization. The superior performance of IMPQ, particularly in the sub-4-bit regime where inter-layer error propagation becomes most severe, confirms our central hypothesis: modeling quantization as a cooperative game that accounts for layer interactions is critical. This marks a conceptual shift from viewing layers as independent entities to understanding them as interconnected components whose collective behavior dictates the final performance of the quantized model.

The primary limitation of our approach is the computational overhead associated with the SPQE-based Shapley value estimation. For a model like Llama-3.1-8B, this process requires approximately 18 hours on a single A40 GPU. However, we argue that this is a practical trade-off. The cost is a one-time analysis, which is then amortized across many deployments of the resulting highly-optimized model. Furthermore, as our ablation study indicates, meaningful importance signals emerge with relatively few SPQE samples, suggesting avenues for reducing this initial cost without catastrophic loss in quality.

This research opens several promising directions for future work. First, the efficiency of the Shapley estimation could be improved by exploring more advanced sampling techniques beyond standard Monte Carlo, such as stratified Monte Carlo sampling, which may converge faster. Second, the interaction-aware framework of IMPQ is not limited to quantization; its principles could be extended to other structured compression techniques, such as layer or head pruning, where component interdependencies are equally critical. Finally, exploring a more granular set of precision assignments beyond the binary 2/4-bit choice could yield further performance gains, although this would increase the complexity of the optimization problem.

## 7 CONCLUSION

In this work, we demonstrate that modeling inter-layer dependencies is critical for effective low-bit LLM quantization. To the best of our knowledge, we are the first to formalize mixed-precision quantization as a cooperative game among layers. Our proposed framework, IMPQ, introduces Shapley-based progressive estimation (SPQE) to capture interaction effects and formulates the bit allocation as a solvable MILP. Comprehensive experiments show IMPQ consistently outperforming prior methods across diverse models (Llama-3, Gemma-2, Qwen-3) and PTQ backends. The framework achieves a significant perplexity reduction of **20% to 80%** over the strongest baselines, particularly as the bit-width tightens.

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

## A  APPENDIX

## EFFECT OF DIAGONAL SHRINKAGE ON IMPQ

| Model | $\alpha = 0.0$ | $\alpha = 0.5$ | $\alpha = 1.0$ |
|---|---|---|---|
| Llama 3.2-3B | $2.83 \times 10^3$ | $2.79 \times 10^3$ | $2.81 \times 10^3$ |

Table 3: Average Perplexity in the range of 2-bit and 4-bit across different alpha values for Llama 3.2-3B on Quanto.

We ablate the diagonal shrinkage hyperparameter $\alpha$ shown in Eq. 9, which balances preserving cross-layer interactions - low $\alpha$ against suppressing off-diagonal noise - high $\alpha$. On Llama 3.2-3B, an intermediate value of $\alpha = 0.5$ achieves the optimal perplexity of $2.79 \times 10^3$ as shown in Table 3. This outperforms both using the full, noisy covariance matrix where $\alpha = 0.0$ with Perplexiy $2.83 \times 10^3$ and completely ignoring interactions where $\alpha = 1.0$ with Perplexity $2.81 \times 10^3$. This result validates our core hypothesis: the off-diagonal terms contain both valuable interaction signals, making $\alpha = 0.5$ and $\alpha = 1.0$ sub-optimal. Our shrinkage approach thus effectively filters this noise while retaining essential interaction data, leading to more robust quantization results.

## BASELINES

### Z-SCORE BASELINE DESCRIPTION

The Z-score baseline, introduced by Dumitru et al. (2024b), provides a data-free approach for measuring layer importance in transformer models. For a given layer $L_i$, the Z-score distribution (ZD) examines the proportion of weights that exhibit values significantly different from the mean. Specifically, it calculates the ratio of weights whose z-scores exceed 1, where the z-score for a weight $w$ is defined as:

$$z = \frac{w - \mu}{\sigma}$$

Here, $\mu$ represents the mean of all weights in the layer and $\sigma$ their standard deviation. The final ZD score for layer $L_i$ is computed as:

$$\text{ZD}(L_i) = \frac{|\{w \in L_i : z(w) > 1\}|}{|L_i|}$$

where $|L_i|$ denotes the total number of weights in layer $i$. This metric assumes that layers with more outlier weights (those deviating significantly from the mean) are more important for the model's functionality. A key advantage of this approach is that it requires no calibration data, making it particularly efficient for rapid layer importance assessment in large language models.

### LAYER INPUT MODIFICATION (LIM) BASELINE DESCRIPTION

The Layer Input Modification (LIM) baseline, also introduced by Dumitru et al. (2024b), measures how significantly a transformer layer modifies its input representations. Unlike the Z-score approach, LIM requires a calibration dataset. While the original work used PG19 (Rae et al., 2019), in our experiments, we use 1000 samples from the C4 (Colossal Clean Crawled Corpus) training set (Raffel et al., 2020) for fair comparison across all methods and models.

For a given layer $L_i$, LIM computes the negative cosine similarity between the layer's input embeddings $\mathbf{L_i^I}$ and output embeddings $\mathbf{L_i^O}$:

$$\text{LIM}(L_i) = -\frac{\mathbf{L_i^I} \cdot \mathbf{L_i^O}}{\|\mathbf{L_i^I}\|\|\mathbf{L_i^O}\|}$$

The intuition behind this metric is that layers that substantially transform their inputs (resulting in low cosine similarity and thus a high negative score) are more important for the model's function than layers that make minimal modifications to their inputs. The negative sign ensures that more important layers receive higher positive scores.

### LLM-MQ SENSITIVITY SCORING DESCRIPTION

LLM-MQ (Li et al., 2023a) introduces a sensitivity-based precision allocation method that uses first-order Taylor approximation to determine how sensitive each layer is to quantization. For a given layer $i$ with weights $\mathbf{W}_i$, the method estimates how quantizing the weights to $b$ bits (denoted by quantization function $Q_b$) affects the model's loss function $\mathcal{L}$:

$$\mathcal{L}(Q_b(\mathbf{W}_i)) \approx \mathcal{L}(\mathbf{W}) + \mathbf{g}_i^T(\mathbf{W}_i - Q_b(\mathbf{W}_i))$$

where $\mathbf{g}_i$ is the gradient of the loss function with respect to the weights of layer $i$. The sensitivity score $s_{i,b}$ for layer $i$ at bit-width $b$ is then computed as:

$$s_{i,b} = |\mathbf{g}_i^T(\mathbf{W}_i - Q_b(\mathbf{W}_i))|$$

This score captures how much the quantization of a layer's weights is expected to impact the model's performance. A higher score indicates that the layer is more sensitive to quantization and thus should be allocated more bits to preserve model performance. The bit allocation is formulated as an integer programming problem that minimizes the sum of sensitivity scores across all layers while respecting memory budget constraints:

$$\arg\min_{c_{i,b}} \sum_i^N \sum_b c_{i,b} \cdot s_{i,b}$$

$$\text{s.t. } \sum_b c_{i,b} = 1, \quad \sum_i^N \sum_b c_{i,b} \cdot \mathcal{M}(Q_b(\mathbf{W}_i)) \leq \mathcal{B}$$

$$c_{i,b} \in \{0,1\}, b \in \{2,3,4\}$$

where $c_{i,b}$ is a binary indicator for whether layer $i$ should use $b$ bits, $\mathcal{M}$ calculates memory usage, and $\mathcal{B}$ is the target memory budget. This formulation allows LLM-MQ to find a bit allocation that minimizes performance degradation while meeting memory constraints.

### ACTIVATION-BASED SCORING DESCRIPTION

Activation-based scoring (Kong et al.) assesses layer importance by calculating the Frobenius norm of layer activations. For a given layer $l$ with hidden states $\mathbf{H}^{(l)}$ of shape $(B, T, D)$ where $B$ is batch size, $T$ is sequence length, and $D$ is hidden dimension, the Frobenius norm is computed as:

$$\|\mathbf{H}^{(l)}\|_F = \sqrt{\sum_{b=1}^{B} \sum_{t=1}^{T} M_{b,t} \sum_{k=1}^{D} \left(\mathbf{H}_{b,t,k}^{(l)}\right)^2}$$

where $M_{b,t}$ is the attention mask (1 for real tokens, 0 for padding). The importance score for layer $i$ is computed relative to the minimum norm across all layers:

$$s_i = 100 \times \frac{\min_j \|\mathbf{H}^{(j)}\|_F}{\|\mathbf{H}^{(i)}\|_F}$$

### RESULT VISUALIZATIONS ON QUANTO AND HQQ

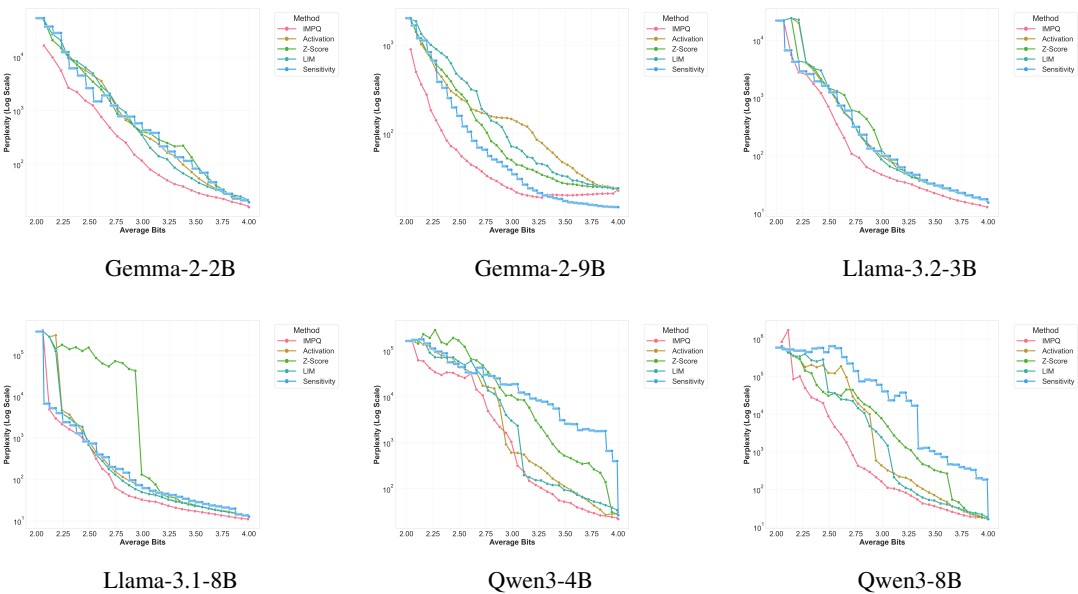

Figure 3: Wikitext-2 Perplexity comparison of quantization methods across Gemma, Llama, Qwen models on HQQ.

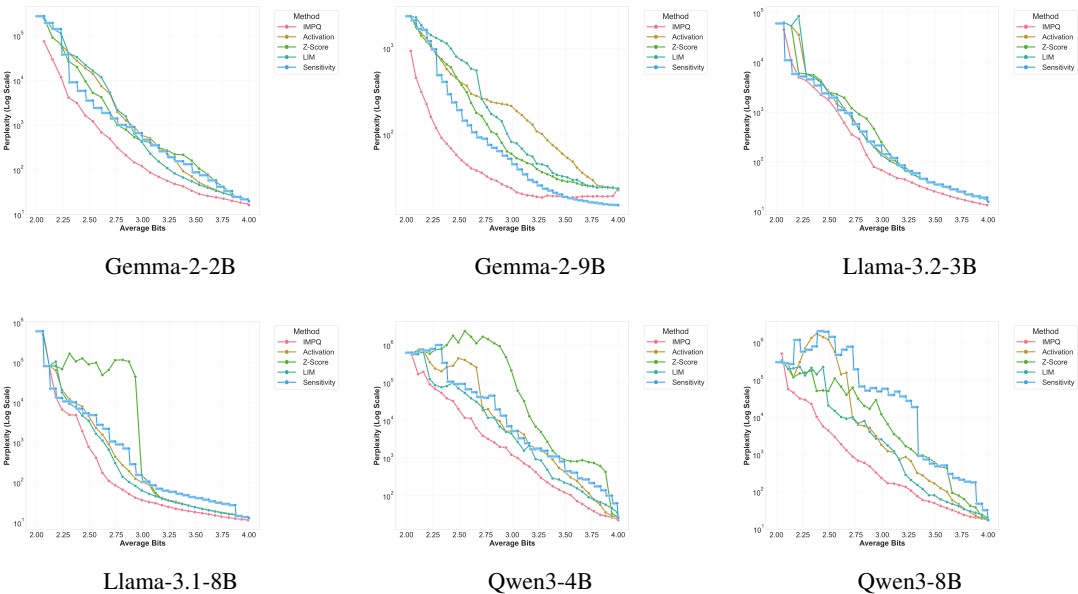

Figure 4: Wikitext-2 Perplexity comparison of quantization methods across Gemma, Llama, Qwen models on Quanto.

