# OpenReview forum: "IMPQ: Interaction-Aware Layerwise Mixed Precision Quantization for LLMs"
_ICLR.cc/2026/Conference — ICLR 2026 Conference Desk Rejected Submission_

### Official Review · Reviewer_bM4D · 2025-10-27

**Soundness:** 3
**Presentation:** 3
**Contribution:** 2
**Rating:** 6
**Confidence:** 4

**Summary:**

This paper introduces IMPQ, a novel framework for mixed-precision quantization of Large Language Models (LLMs) that addresses the critical challenge of deploying massive models on resource-constrained devices. The core innovation lies in modeling quantization as a cooperative game among transformer layers, where the authors propose SPQE (Shapley-based Progressive Quantization Estimation) to capture layer sensitivities and inter-layer interactions through progressive quantization rather than abrupt pruning.

**Strengths:**

The application of cooperative game theory and Shapley values to mixed-precision quantization represents a significant conceptual advance. By framing layers as players in a cooperative game, the authors provide a principled approach to quantifying both individual layer contributions and inter-layer interactions, which existing methods neglect.

SPQE's progressive quantization (from 4-bit to 2-bit) is a clever innovation that maintains model stability during Shapley estimation. This approach effectively avoids the catastrophic performance degradation and high variance associated with layer pruning, enabling more accurate and reliable layer importance assessment.

**Weaknesses:**

While the paper analyzes the impact of Monte Carlo samples, it neglects a thorough examination of other hyperparameters. The diagonal shrinkage parameter $\alpha$ is fixed at 0.5 without justification or sensitivity analysis. The choice of baseline (4-bit) and target (2-bit) precisions is also not motivated or varied.

Experiments rely primarily on C4 for Shapley estimation and WikiText-2 for evaluation. Testing on more diverse domains (e.g., code, multilingual text) and larger datasets would strengthen claims about generalizability, especially given the domain-specific nature of quantization effects.
While several baselines are included, comparisons with recent Hessian-based methods like HAWQ are limited. The paper also doesn't compare against neural architecture search approaches for quantization, which could provide additional context for the performance gains.


I'm mainly interested in the presentation and theoretical parts, with some confusing content below. Please tell me if I was wrong.
The paper assumes Monte Carlo sampling provides accurate Shapley approximations without a theoretical analysis of approximation error. The value function $v_{\text{NLL}}(S) = \mathbb{E}_{(x,t)\sim D}[-\log p(x_{t+1}|x_{\leq t}; S)]$ in Equation 3 is not justified as the optimal choice for measuring layer contributions in the cooperative game framework.

Section 3.2 begins with a second-order Taylor approximation $\Delta L \approx \sum_{i=1}^{L} g_i^\top \epsilon_i + \sum_{i=1}^{L} \sum_{j=1}^{L} \epsilon_i^\top H_{ij} \epsilon_j$ but then switches to a Shapley-based approach without reconciling these perspectives. The covariance matrix $C = \frac{1}{M}(\Delta v_\ell - \hat{\phi})^\top (\Delta v_\ell - \hat{\phi})$ in Equation 8 is proposed as a Hessian proxy without theoretical justification for this equivalence.

The distribution $D$ in Equation 3 is not clearly specified. The perturbation $\epsilon_i$ in Section 3.2 is used without defining its relationship to quantization error. The construction of the covariance matrix $C$ in Equation 8 lacks clarity regarding dimensions and the precise meaning of $\Delta v_\ell$. It's kindly suggested to check these notations.

**Questions:**

See the weakness, mainly in the exp settings, theory, and notations.

---

> ### Author Response · Authors · 2025-11-19
> **Response to Review bM4D Part 1**
>
> We thank the reviewer bM4D for the thoughtful assessment and for recognizing the conceptual advance of applying cooperative game theory to quantization, as well as the innovation of SPQE in maintaining model stability! We will address the concerns in the weakness section below.
>
> $\textbf{Response to Hyperparameter}$ $\alpha$:
>
> We selected $\alpha=0.5$ based on the ablation in  Appendix Table 3. For Llama-3.2-3B, $\alpha=0.5$ achieves the perplexity $2.79 \times 10^3$, outperforming both $\alpha=0.0$, $2.83 \times 10^3$ and the $\alpha=1.0$, $2.81 \times 10^3$.
>
> Crucially, while $\alpha=0.5$ is optimal, the performance curve is relatively flat, indicating that our method is robust and not sensitive to hyperparameter tuning as the Shapley values provide the dominant signal.
>
> $\textbf{Response to 2/4 bit}$:
>
> While our experiments focused on the 2/4-bit regime to demonstrate efficacy under strict compression, the IMPQ framework is not theoretically limited to binary choices. Specifically, the current binary decision variable $q_i \in \\{0, 1\\}$ (Eq 11 - 13) can be straightforwardly generalized to a multi-choice variable (e.g. using integer variables or one-hot encoding) representing a set of available bit-widths $\\{2, 3, 4, 8\\}$.
>
> Although this increases the number of decision variables, the underlying objective function remains a quadratic interaction model. The solver simply optimizes over a larger discrete search space. We plan to explore these multi-bit configurations in future work to demonstrate this flexibility.
>
> $\textbf{Response to downstream task evaluations}:$
>
> We acknowledge the reviewer’s comment that downstream evaluation is often desirable. Due to the substantial computational cost of running SPQE across 3 model families with 2 models from each model family, the 3 PTQ backends for every transformer layer quantization, and very limited computational resources, we were unable to run the full suite of downstream benchmarks. However, given the strong correlation between PPL and downstream performance, we believe the significant PPL reductions demonstrate the effectiveness of our method.

---

> > ### Author Response · Authors · 2025-11-19
> > **Response to Review bM4D Part 2**
> >
> > $\textbf{Response to Taylor Expansion to Quadratic Game}$
> >
> > Consider the loss degradation $\Delta L$ under a mixed-precision configuration $b \in \{0,1\}^L$. A second-order Taylor expansion of the loss gives:
> >
> > $\Delta L(b) \approx \sum_i g_i^\top \varepsilon_i(b_i) + \frac{1}{2} \sum_{i,j} \varepsilon_i(b_i)^\top H_{ij} \varepsilon_j(b_j)$
> >
> > Under unbiased quantization noise assumption, the expected loss simplifies to a quadratic pseudo-Boolean function:
> >
> > $$\mathbb{E}[\Delta L(b)] = v(b) \approx \sum_i a_i b_i + \sum_{i<j} a_{ij} b_i b_j$$
> >
> > where $a_i$ represents the isolated sensitivity of layer $i$, and $a_{ij}$ represents the pairwise interaction cost derived from Hessian blocks $H_{ij}$. The goal is to minimize this objective.
> >
> > For the baselines, these methods essentially estimate only the first term ($a_i$) using local proxies. They optimize assuming layers are independent ($a_{ij} \approx 0$) because computing the full Hessian is infeasible. This ignores error propagation, which is critical in low-bit quantization settings.
> >
> > IMPQ aims to minimize the full quadratic objective without computing the Hessian. We use SPQE to estimate the Shapley value $\phi_i$. For the quadratic game defined above, the Shapley value can be derived via Harsanyi dividends [1][2].
> >
> > Specifically, a Harsanyi dividend $\Delta_v(T)$ represents the value uniquely generated by coalition $T$. The Shapley value is defined as the equal sharing of these dividends among the contributors:
> >
> > $$\phi_i(v) = \sum_{T \subseteq N: i \in T} \frac{\Delta_v(T)}{|T|}$$
> >
> > We compute the dividends for our quadratic game $v(b)$ where we identifying $b$ with the set of quantized layers $S=(i | b_i=1)$:
> >
> > For $|T|=1$ (T={i}), by definition, $\Delta_v({i}) = v({i}) = a_i$.
> >
> > For $|T|=2$ (T={i,j}), since $v({i,j}) = a_i + a_j + a_{ij}$ and by definition $v({i,j}) = \Delta_v({i}) + \Delta_v({j}) + \Delta_v({i,j})$, it follows that $\Delta_v({i,j}) = a_{ij}$.
> >
> > For $|T|>2$, a game defined by a polynomial of degree $k=2$, all higher-order dividends are zero.
> >
> > So, $$\phi_i(v) = \Delta_v({i}) + \sum_{j \neq i} \frac{\Delta_v({i,j})}{2} = a_i + \frac{1}{2} \sum_{j \neq i} a_{ij}$$
> >
> > where the first term captures isolated sensitivity where $|T|=1$, the second captures pairwise interactions where $|T|=2$, and higher-order terms $|T|>2$ vanish in the quadratic approximation.
> >
> > Based on the proof, the Shapley value aggregates both isolated sensitivity and interactions from the Taylor expansion.
> >
> > [1]: A simplified bargaining model for the n-person cooperative game.
> >
> > [2]: An axiomatic approach to the concept of interaction among players in cooperative games.

---

> ### Author Response · Authors · 2025-11-19
> **Response to Review bM4D Part 3**
>
> $\textbf{Response to Question on NLL as payoff}$
>
> We choose NLL as the payoff because the primary goal is to minimize the degradation in language modeling capability, measured directly by PPL. Since $\text{PPL} = \exp(\text{NLL})$, minimizing NLL is monotonically equivalent to minimizing PPL.
>
> At the same time, NLL allows us to leverage the additive axiom of Shapley values, which guarantees additive decomposition of the total potential loss degradation:
>
> $$\sum_{i=1}^L \phi_i = v(\text{All Quantized}) - v(\text{None Quantized}) = \Delta \text{NLL}_{total}$$
>
> This property ensures that $\phi_i$ represents an additive decomposition of the model's total loss degradation. Minimizing the sum of the selected layers' Shapley values is therefore equivalent to minimizing the total NLL increase.
>
> Finally, NLL is a smooth, fine-grained metric, which crucial for the stability of SPQE, which relies on progressive quantization to ensure gradual performance degradation and low variance estimates.
>
> For the MC approximation error, the convergence of MC approximation for Shapley values is well-established, with approximation error decreasing at a rate of $O(1/\sqrt{M})$ . Empirically, our ablation study in Table 2 confirms rapid convergence, in which 50 samples achieve nearly the same perplexity reduction -19.93% Rel. $\Delta$ Geo. Mean as 100 samples -19.88%. This indicates that the approximation error is well controlled in practice with $M=100$.
>
> $\textbf{Response to Question on Covariance}$
>
> While the Shapley value $\phi_i$ inherently aggregates interaction effects proven in part 2, using it directly (i.e., $\alpha=1.0$) restricts us to linear optimization ($\min \sum \phi_i q_i$). This treats a layer's interaction cost as a fixed penalty, regardless of which other layers are quantized.
>
> Intuitively, if two layers interact strongly large $a_{ij}$, their marginal contributions will covary significantly depending on the quantization order.
>
> Therefore, we choose to formulate the problem as quadratic optimization ($\min a^\top q + q^\top K q$), we need to disentangle the pairwise term $a_{ij}$ from the isolated sensitivity $a_i$. We use the covariance matrix $C$ of the marginal contributions as a proxy to reconstruct this.
>
> $\textbf{Response to Question on Notations}$
>
> We apologize for the lack of clarity in the initial submission and will revise the notation as follows.
>
> 1. $\mathcal{D}$ in Eq. 3: $\mathcal{D}$ represents the calibration corpus, specifically the C4 training dataset.
>
> 2. The notation $(x, t) \sim \mathcal{D}$ refers to sampling a sequence $x$ and token position $t$ from this corpus.
>
> 3. $\epsilon_i$ in section 3.2: The perturbation $\epsilon_i$ represents the quantization error in layer $i$'s weights. Specifically, $\epsilon_i = Q(W_i) - W_i$, where $Q(\cdot)$ is the quantization function. This leads to the perturbed weights $\tilde{W}_i = W_i + \epsilon_i$ used in the Taylor expansion.
>
> 4. Covariance Matrix $C$ in Eq. 8: We clarify the dimension and construction of $C$. Let $V \in \mathbb{R}^{M \times L}$ be the matrix where $V_{m,i}$ is the marginal contribution of layer $i$ in permutation $m$.
>
> 5. Let $\hat{\phi} \in \mathbb{R}^{1 \times L}$ be the vector of estimated Shapley values (column means of $V$). The covariance matrix $C \in \mathbb{R}^{L \times L}$ is:$$C = \frac{1}{M} (V - \mathbf{1}\hat{\phi})^\top (V - \mathbf{1}\hat{\phi})$$where $\mathbf{1}$ is a column vector of ones of size $M$.

---

### Official Review · Reviewer_GnxT · 2025-10-31

**Soundness:** 4
**Presentation:** 3
**Contribution:** 3
**Rating:** 6
**Confidence:** 4

**Summary:**

This paper introduces IMPQ, a mixed precision PTQ method for LLMs that leverages Shapley values to estimate the importance/sensitivity of individual layers in LLMs, and forms a Hessian scaled objective for the mixed-precision problem that can be solved using quadratic integer programming solvers.

**Strengths:**

The method is clear and the evaluation supports its claim of effectiveness over baselines.

**Weaknesses:**

- The complexity of the algorithm needs to be analyzed: The calculation of Shapley value and the Monte Carlo samping seems very computationally expensive for LLMs. A comparison of the computational complexity of the IMPQ method against baselines may be needed given important factors like layer numbers, and sampling numbers
- Downstream task evaluation is missing. The evaluation section only shows result on wikitext perplexity, but IMPQ method and some baselines/quantization methods needs calibration. It will be more convincible if results on downstream tasks are included. Given this perplexity, how hard it  will be to apply to MoE models?

**Questions:**

Besides the weakness section, could the author answer the following questions

1. Why the average per-token NLL are used as pay-offs?
2. Could the author explain/give some intuitive hints on why IMPQ  outperforms baselines?
3. Apart from the computationaly complexity analysis in Weakness section, could the author give a comparison of quantization time? This will be straightforward for readers to understand the complexity-performance trane off
4. The selection of hyper params like $\alpha$ in line 260. Could the author explain why choose this specific value of 0.5?

There are a few typos in the paper due to citation formating issue like line 109 and 112

---

> ### Author Response · Authors · 2025-11-19
> **Response to Review GnxT Part 1**
>
> We thank Reviewer GnxT for their thorough review and positive assessment of our work. We address the questions regarding complexity, downstream tasks, and intuition below.
>
> $\textbf{Response to W1}:$
>
> We acknowledge that Shapley value estimation via SPQE is computationally intensive compared to baseline heuristics.
>
> Let $L$ be the number of layers, $M$ be the number of Monte Carlo samples, and $T_{fwd}$ be the time for a forward pass on the calibration set.
>
> Our SPQE complexity is $O(M \cdot L \cdot T_{fwd})$. We iterate through $M$ permutations, and for each permutation, we perform $L$ sequential quantization steps, requiring a forward pass evaluation at each step to measure the marginal contribution (Section 3.1).
>
> For the baselines, these typically require $O(T_{fwd})$ needing only one pass to gather layerwise statistics.
>
> We acknowledge that SPQE cost is substantial, but we emphasize that it is a one-time, offline analysis cost amortized across all future deployments of the model. The result is a 20-80% perplexity reduction (Table 1), which often justifies this initial investment. Furthermore, our ablation (Table 2) demonstrates that reducing samples to $M=50$ halves the runtime (~9 hours) with minimal quality loss (-19.93% vs -19.88% relative gain) 9, making the method more practical.
>
> $\textbf{Response to W2}:$
>
> We acknowledge the reviewer’s comment that downstream evaluation is often desirable. Due to the substantial computational cost of running SPQE across 3 model families with 2 models from each model family, the 3 PTQ backends for every transformer layer quantization, and very limited computational resources, we were unable to run the full suite of downstream benchmarks. However, given the strong correlation between PPL and downstream performance in the paper, we believe the significant PPL reductions demonstrate the effectiveness of our method.

---

> > ### Author Response · Authors · 2025-11-19
> > **Response to Review GnxT Part 2**
> >
> > $\textbf{Response to IMPQ intuition}:$
> >
> > The intuition that IMPQ outperforms baselines is its ability to capture inter-layer interactions, which existing methods largely ignore. We can demonstrate this via a second-order analysis.
> >
> > Consider the loss degradation $\Delta L$ under a mixed-precision configuration $b \in \\{0,1\\}^L$. A second-order Taylor expansion of the loss gives:
> >
> > $\Delta L(b) \approx \sum_i g_i^\top \varepsilon_i(b_i) + \frac{1}{2} \sum_{i,j} \varepsilon_i(b_i)^\top H_{ij} \varepsilon_j(b_j)$
> >
> > Under unbiased quantization noise assumption, the expected loss simplifies to a quadratic pseudo-Boolean function:
> >
> > $$\mathbb{E}[\Delta L(b)] = v(b) \approx \sum_i a_i b_i + \sum_{i<j} a_{ij} b_i b_j$$
> >
> > where $a_i$ represents the isolated sensitivity of layer $i$, and $a_{ij}$ represents the pairwise interaction cost derived from Hessian blocks $H_{ij}$. The goal is to minimize this objective.
> >
> > For the baselines, these methods essentially estimate only the first term ($a_i$) using local proxies. They optimize assuming layers are independent ($a_{ij} \approx 0$) because computing the full Hessian is infeasible. This ignores error propagation, which is critical in low-bit quantization settings.
> >
> > IMPQ aims to minimize the full quadratic objective without computing the Hessian. We use SPQE to estimate the Shapley value $\phi_i$. For the quadratic game defined above, the Shapley value can be derived via Harsanyi dividends [1][2].
> >
> > Specifically, a Harsanyi dividend $\Delta_v(T)$ represents the value uniquely generated by coalition $T$. The Shapley value is defined as the equal sharing of these dividends among the contributors:
> >
> > $$\phi_i(v) = \sum_{T \subseteq N: i \in T} \frac{\Delta_v(T)}{|T|}$$
> >
> > We compute the dividends for our quadratic game $v(b)$ where we identifying $b$ with the set of quantized layers $S=(i | b_i=1)$:
> >
> > For $|T|=1$ (T={i}), by definition, $\Delta_v(\{i\}) = v(\{i\}) = a_i$.
> >
> > For $|T|=2$ (T={i,j}), since $v(\{i,j\}) = a_i + a_j + a_{ij}$ and by definition $v(\{i,j\}) = \Delta_v(\{i\}) + \Delta_v(\{j\}) + \Delta_v(\{i,j\})$, it follows that $\Delta_v(\{i,j\}) = a_{ij}$.
> >
> > For $|T|>2$, a game defined by a polynomial of degree $k=2$, all higher-order dividends are zero.
> >
> > So, $$\phi_i(v) = \Delta_v(\{i\}) + \sum_{j \neq i} \frac{\Delta_v(\{i,j\})}{2} =  a_i + \frac{1}{2} \sum_{j \neq i} a_{ij}$$
> >
> > where the first term captures isolated sensitivity where $|T|=1$, the second captures pairwise interactions where $|T|=2$, and higher-order terms $|T|>2$ vanish in the quadratic approximation.
> >
> > Based on the proof, the Shapley value aggregates both isolated sensitivity and interactions.
> >
> > Our method IMPQ combines these estimates (Eq 8-10) to reconstruct and solve the quadratic optimization problem. This  optimization minimizes the global impact of error propagation, leading to superior performance. We will including the full proof in our final version paper appendix to show further clarification.
> >
> > [1]: A simplified bargaining model for the n-person cooperative game.
> >
> > [2]: An axiomatic approach to the concept of interaction among players in cooperative games.

---

> > > ### Author Response · Authors · 2025-11-19
> > > **Response to Review GnxT Part 3**
> > >
> > > $\textbf{Response to NLL}:$
> > >
> > > We choose NLL as the payoff because the primary goal is to minimize the degradation in language modeling capability, measured directly by PPL. Since $\text{PPL} = \exp(\text{NLL})$, minimizing NLL is monotonically equivalent to minimizing PPL.
> > >
> > > At the same time, NLL allows us to leverage the additive axiom of Shapley values, which guarantees additive decomposition of the total potential loss degradation:
> > >
> > > $$\sum_{i=1}^L \phi_i = v(\text{All Quantized}) - v(\text{None Quantized}) = \Delta \text{NLL}_{total}$$
> > >
> > > This property ensures that $\phi_i$ represents an additive decomposition of the model's total loss degradation. Minimizing the sum of the selected layers' Shapley values is therefore equivalent to minimizing the total NLL increase.
> > >
> > > Finally, NLL is a smooth, fine-grained metric, which crucial for the stability of SPQE, which relies on progressive quantization to ensure gradual performance degradation and low variance estimates.
> > >
> > > $\textbf{Response to MoE}:$
> > >
> > > IMPQ is applicable to MoE models. The complexity depends on the definition of the players.
> > >
> > > If we decide the player to be layerwise, which treats an entire MoE block as a single player, it maintains the same complexity as dense models.
> > >
> > > If we decide the player to be expert-wise, we need to treat each expert as player. In this case, it captures the heterogeneity but increases the number of players, where $$\hat{L} = L_{dense} + L_{MoE} \cdot N_{experts}$$
> > >
> > > While this increases the SPQE runtime complexity linearly, from  $O(M \cdot L \cdot T_{fwd})$ to  $O(M \cdot \hat{L} \cdot T_{fwd})$.
> > >
> > > $\textbf{Response to Quantization Time}:$
> > >
> > > The actual quantization time is identical for all methods once the bit allocation is determined. We will provide the quantization time estimate in our final version paper appendix.
> > >
> > > $\textbf{Response to Hyperparameter}$ $\alpha$:
> > >
> > > We selected $\alpha=0.5$ based on the ablation in  Appendix Table 3. For Llama-3.2-3B, $\alpha=0.5$ achieves the perplexity $2.79 \times 10^3$, outperforming both $\alpha=0.0$, $2.83 \times 10^3$ and the $\alpha=1.0$, $2.81 \times 10^3$.
> > >
> > > Crucially, while $\alpha=0.5$ is optimal, the performance curve is relatively flat, indicating that our method is robust and not sensitive to hyperparameter tuning as the Shapley values provide the dominant signal.

---

> > > > ### Comment · Reviewer_GnxT · 2025-11-27
> > > >
> > > > Thanks for clarification. Please add the additional analysis to the manuscript. I remain positive about this paper, and adjusted my scores accordingly.

---

> > > > > ### Author Response · Authors · 2025-11-27
> > > > > **Close Statement**
> > > > >
> > > > > Dear Reviewer GnxT, we sincerely appreciate your positive feedback on our work. I will make sure to cover the additional analysis in our final version paper.
> > > > >
> > > > > Thank you so much!

---

### Official Review · Reviewer_TvGU · 2025-11-01

**Soundness:** 3
**Presentation:** 2
**Contribution:** 3
**Rating:** 4
**Confidence:** 5

**Summary:**

This paper introduces SPQE, a Shapley-based approach for estimating layer importance via progressive quantization, and IMPQ, a MILP-based method for assigning 2- or 4-bit precision under memory constraints. The authors frame mixed-precision quantization as a cooperative game among layers, capturing inter-layer dependencies more effectively than existing heuristics. Evaluated on several LLMs and PTQ backends, IMPQ achieves significantly lower perplexity, especially under 2-bit constraints, demonstrating strong empirical performance and robustness across models and settings.

**Strengths:**

The key strengths lie in the originality of modeling quantization as a cooperative game, the method’s stability under aggressive bit reductions, and the extensive and thorough experimental validation. The results clearly show that accounting for inter-layer interactions leads to better bit allocation and quantized performance than isolated sensitivity measures.

**Weaknesses:**

1. The method carries substantial computational overhead, with SPQE requiring many hours to estimate Shapley values even for mid-sized models.

2. The approach is currently limited to binary 2-bit/4-bit decisions, which restricts its generality, and the MILP formulation, though optimal in theory, raises questions about scalability to larger models or finer-grained bit options.

3. Moreover, the paper does not explore how robust the final assignments are to noise in Shapley estimates, nor does it fully explain implementation details such as memory constraint handling or solver configurations.

**Questions:**

1. It would help to know whether the authors plan to support finer bit precision (e.g., 3-bit or 8-bit layers), and whether SPQE or MILP runtimes could be reduced through approximation or more scalable formulations.

2. Additionally, can this game-theoretic framework extend to other compression tasks, such as structured pruning, where modeling interactions is equally important?

---

> ### Author Response · Authors · 2025-11-19
> **Response to Review TvGU Part 1**
>
> We sincerely thank Reviewer TvGU for their insightful review and positive assessment of our work. We are encouraged that the reviewer recognized the originality of modeling quantization as a cooperative game, the method’s stability under aggressive bit reductions, and the extensive experimental validation. We appreciate the constructive feedback and address the concerns regarding computational overhead, generality, scalability, robustness, and implementation details below.
>
> $\textbf{Response to W1}:$
>
> We acknowledge the computational overhead associated with SPQE. As noted in the Discussion (Lines 461-463), estimating Shapley values for Llama-3.1-8B required approximately 18 hours on a single A40 GPU using 100 Monte Carlo samples. However, we emphasize that this is a one-time, offline analysis cost that is amortized across the numerous deployments of the resulting optimized model (Lines 463-465). The significant 20%-80% relative perplexity reductions achieved by IMPQ often justify this initial investment.
>
> Furthermore, our ablation study (Section 5, Table 2) demonstrates that SPQE converges rapidly. We observe that 50 samples achieve nearly the same perplexity improvement (Rel. $\Delta$ Geo. Mean of -19.93%) as 100 samples (-19.88%). This indicates that the runtime can be nearly halved with minimal impact on the final quantization quality.
>
> $\textbf{Response to W2}:$
>
> We appreciate the reviewer's questions regarding the flexibility and scalability of our optimization framework. We address both aspects below.
>
> First, while our experiments focused on the challenging 2/4-bit regime to demonstrate efficacy under strict compression, the IMPQ framework is not theoretically limited to binary choices. Specifically, the current binary decision variable $q_i \in \\{0, 1\\}$ (Eq 11 - 13) can be straightforwardly generalized to a multi-choice variable (e.g. using integer variables or one-hot encoding) representing a set of available bit-widths $\\{2, 3, 4, 8\\}$.
>
> Although this increases the number of decision variables, the underlying objective function remains a quadratic interaction model. The solver simply optimizes over a larger discrete search space. We plan to explore these multi-bit configurations in future work to demonstrate this flexibility.
>
> Second, we clarify that the computational complexity of our MILP depends on the number of layers, not the total parameter count. Even for larger models like Llama-3-70B, the number of layers remains relatively small in the context of integer programming. Standard solvers can solve binary quadratic problems of this size to global optimality in seconds. The primary computational cost is the offline Shapley estimation (SPQE), not the MILP solving. Therefore, the optimization step itself is highly scalable to substantially larger models and finer-grained bit options without becoming a computational bottleneck.

---

> > ### Author Response · Authors · 2025-11-19
> > **Response to Review TvGU Part 2**
> >
> > $\textbf{Response to W3}:$
> >
> > $\textbf{Regarding the robustness to estimation noise}$, our results demonstrate that IMPQ is highly robust to noise for two key reasons.
> >
> > First, we have demonstrated Shapley estimation stability via SPQE. Unlike pruning-based methods that cause catastrophic degradation with high-variance, SPQE uses progressive quantization. This ensures smooth performance changes, leading to low-variance, stable Shapley estimates (see Figure 2).
> >
> > Second, we explicitly tested robustness to noise via our diagonal shrinkage ablation (Appendix Table 3). We introduced the hyperparameter $\alpha$ to filter noise from the interaction matrix. The results show that the optimal filtered setting $\alpha=0.5$, PPL $2.79\times10^{3}$ performs only marginally better than the raw/noisy setting $\alpha=0.0$, PPL $2.83\times10^{3}$ or the diagonal-only setting $\alpha=1.0$, PPL $2.81\times10^{3}$.
> >
> > The fact that the final perplexity is relatively insensitive to how aggressively we filter the covariance matrix proves that the underlying Shapley signal is strong and the final bit assignment is robust to sampling noise.
> >
> > $\textbf{Regarding the memory constraint details}$, in our setup, each transformer layer is quantized either to 2 bits or 4 bits.
> >
> > We first compute the total memory if all quantizable layers are at 2 bits (call this $B_{low}$) and if all are at 4 bits (call this $B_{high}$). For any target average bit-width $b_{avg}$ between 2 and 4 (for example, 2.25), we define the corresponding memory budget by linear interpolation:
> >
> > $B(b_{avg}) = B_{low} + ((b_{avg} − 2) / (4 − 2)) * (B_{high} − B_{low})$.
> >
> > Intuitively, this budget lies between the all layers 2-bit and all layers 4-bit footprints and corresponds to some mixture of 2-bit and 4-bit layers. IMPQ and all baselines methods (Activation, Sensitivity, LIM, Z-Score) must choose which layers are 2-bit vs. 4-bit subject to the same budget $B(b_{avg})$.
> >
> > $\textbf{Regarding the solver configuration}$, we used the SCIP optimization suite with its default configuration. We found that the standard settings were sufficient to solve our layer-wise binary quadratic problems to optimality in seconds, with no special tuning required.
> >
> > We will explicitly describe this memory-budget interpolation procedure and the solver configuration details in the final version, and they were omitted from the current submission due to the conference page limit.
> >
> > $\textbf{Reponse to Q1}:$
> >
> > Please refer to $\textbf{Response to W2}$.
> >
> > $\textbf{Reponse to Q2}:$
> >
> > Yes, the game-theoretic framework is extensible to other compression tasks. As we mention in the discussion, "its principles could be extended to other structured compression techniques, such as layer or head pruning, where component interdependencies are equally critical" (Lines 472 - 475).

---

### Official Review · Reviewer_8JN3 · 2025-11-02

**Soundness:** 2
**Presentation:** 2
**Contribution:** 2
**Rating:** 2
**Confidence:** 3

**Summary:**

This paper proposes SPQE to obtain accurate Shapley estimates of layer sensitivities and inter-layer interactions. SPQE is based on cooperative game theory. The authors also use IMPQ to find optimal bit-width assignments.

**Strengths:**

1. An innovative use of Shapley value analysis and cooperative games among LLM layers to model mixed-precision quantization.

2. Demonstrated performance improvements on Llama-3, Gemma-2, and Qwen-3.

**Weaknesses:**

1. Why is modeling mixed-precision quantization using Shapley value analysis and cooperative games among LLM layers more effective than traditional Hessian-based methods? The authors did not clearly explain the motivation.

2. The experimental setup seems problematic. I believe the paper does not evaluate quantization performance under an accepted standard setting.
(a) The performance of the full-precision baseline is not stated.
(b) The bit range defined in the paper (e.g., 2.5–3.0) is overly broad. It is not explained how these bits correspond to specific mixing ratios or group sizes, nor how fairness is maintained across different baselines.
(c) The statistical results in the paper are inconsistent with previous work. A perplexity (ppl) around 15–25 is too high—much worse than those reported in existing papers. For example, in [1], methods such as OmniQuant and CherryQ achieve ppl < 10 at 2.15 bits. Although the experimental setup in [1] differs from this paper, the discrepancy should not be this large.

3. Continuing from 2.(c), I believe that while parameter sensitivity has room for optimization, improving only the sensitivity is of limited benefit. The optimal sensitivity selection strategy may only result in a small decrease in perplexity. Therefore, the claim in Table 1 that IMPQ reduces ppl by about 10 compared to other baselines may not be credible. The authors may not have obtained the optimal performance for the baselines.
4. Some writing issues—e.g., the font in Table 1 is too small.

[1] Cherry on Top: Parameter Heterogeneity and Quantization in Large Language Models

**Questions:**

Please see weaknesses.

---

> ### Author Response · Authors · 2025-11-17
> **Response to Review 8JN3: Part 1**
>
> We thank Reviewer 8JN3 for their detailed feedback. We appreciate the opportunity to clarify our motivation and experimental setup, which we recognize were not sufficiently detailed in the initial submission due to the strict page limits. We believe these clarifications will address the reviewer's valid concerns and misunderstandings about our methodology and the credibility of our results. We will incorporate all suggested clarifications and formatting improvements in the final version.
>
> $\textbf{Response to W1}$:
>
> Our method is motivated by a specific limitation of existing Hessian-based mixed-precision PTQ: it is practically impossible to compute the full Hessian for LLMs, so existing methods drop almost all cross-layer interaction terms.
>
> Traditional mixed-precision PTQ assigns bit widths using isolated heuristics or diagonal/block-diagonal Hessian approximations. As we note in the paper, these approaches “assess layer sensitivities independently using heuristics like norm-based metrics or Hessian approximation, neglecting inter-layer dependencies” (Line 116-117).
>
> For a multi-billion-parameter model, the full Hessian would have $O(N^2)$ entries, which is completely infeasible to store and compute for LLMs. In practice, mixed-precision methods therefore rely on layer-wise Hessian statistics. For example, HAWQ [1] uses the top eigenvalue of each layer’s Hessian, and HAWQ-V2 [2] uses the average Hessian trace per layer to rank sensitivity and select layer bit-widths. This effectively treats layers independently and ignores cross-layer Hessian blocks $H_{i,j}​ (i \neq j)$ that describe how quantization noise in one layer interacts with subsequent layers.
>
> Given this issue, our proposed SPQE is designed to recover these interaction effects without ever forming the full Hessian. We model mixed-precision quantization as a cooperative game over layers in which we progressively quantize layers in many random orders and measure the loss change when each layer is quantized in different coalitions.
>
> The resulting Shapley values capture each layer’s global importance, and the covariance of Shapley deviations serves as a low-dimensional proxy for cross-layer interactions (Eq 6-8). In other words, SPQE gives us a data-driven estimate of both first-order sensitivities and second-order interaction information at the layer level, using only a manageable number of forward passes on a calibration set and no explicit Hessian computation.
>
> [1] HAWQ: Hessian AWare Quantization of Neural Networks with Mixed-Precision.
> [2] HAWQ-V2: HAWQ-V2: Hessian Aware trace-Weighted Quantization of Neural Networks.
>
> $\textbf{Response to W2.a}$:
>
> We appreciate the reviewer’s concern and agree that explicitly reporting the full-precision perplexity improves clarity. In the current submission, we only report the perplexities of quantized models under different average bit budgets (Table 1 and the Figure 1, 3, 4), following common PTQ practice of focusing on relative degradation under a fixed evaluation setup. The full-precision model is used as the reference in all experiments, but the perplexity is not explicitly listed due to the space constraints. In our final version, we will make this explicit by adding a full-precision row in Table 1.

---

> ### Author Response · Authors · 2025-11-17
> **Response to Review 8JN3: Part 2**
>
> $\textbf{Response to W2.b}$:
>
> In our setup, each transformer layer is quantized either to 2 bits or 4 bits. We first compute the total memory if all quantizable layers are at 2 bits (call this $B_{low}$) and if all are at 4 bits (call this $B_{high}$). For any target average bit-width $b_{avg}$ between 2 and 4 (for example, 2.25), we define the corresponding memory budget by linear interpolation:
>
> $B(b_{avg}) = B_{low} + ((b_{avg} − 2) / (4 − 2)) * (B_{high} − B_{low})$.
>
> Intuitively, this budget lies between the $\textbf{all layers 2-bit}$ and $\textbf{all layers 4-bit}$ footprints and corresponds to some mixture of 2-bit and 4-bit layers. IMPQ and all baselines methods (Activation, Sensitivity, LIM, Z-Score) must choose which layers are 2-bit vs. 4-bit subject to the same budget $B(b_{avg})$. We will explicitly describe this memory-budget interpolation procedure in the final version, and it was omitted from the current submission due to the conference page limit.
>
> In the experiments, we sweep $b_{avg}$ over a fine grid between 2 and 4 bits, so each method yields full bit vs. PPL curve for different models and PTQ backends, visualized in Figures 1, 3, 4. The “bit ranges” reported in Table 1 (e.g. 2.5–3.0) are simply bins over these curves: for a range such as 2.5–3.0, we gather all evaluated models whose achieved average bit-width falls in that interval and report the average perplexity over those models (for example, if there are 6 points in 2.5-3.0, we report $(PPL_{1} + … + PPL_{6} )/ 6$.
>
> Because each model has many layers and we evaluate at many intermediate budgets, it is infeasible to list every perplexity vs. bit budget pair in a table within the conference page limit. This is why we show the full bit vs. PPL trade-off in visualizations and only report aggregated averages in Table 1.
>
> Regarding fairness, for each quantization backend (GPTQ, HQQ, Quanto) we use the fixed default group size, calibration data, and all implementation details identically for IMPQ and all baselines. The budgeting and binning procedure described above is applied in exactly the same way to every method; the only difference is how they rank layers when deciding which ones receive 4 bits vs. 2 bits under the common budget B($b_{avg}$). This ensures that all methods are compared under the same memory constraints and quantization configuration.

---

> > ### Author Response · Authors · 2025-11-17
> > **Response to Review 8JN3: Part 3**
> >
> > $\textbf{Response to W2.c and W3}$:
> >
> > We respectfully disagree that our perplexity values are inconsistent with prior work. The comparison to CherryQ is not in a comparable setting. CherryQ operates on a different model family and dataset and allows additional training, whereas our experiments are in a stricter pure post-training quantization regime.
> >
> > In our work, we quantize pretrained Llama-3, Gemma-2, and Qwen-3 models to mixed 2/4-bit without any quantization-aware fine-tuning, and all quantizable parameters in the selected layers are in low precision under the given bit budget. In contrast, CherryQ explicitly uses quantization-aware training and retains a small subset of cherry parameters in FP16 while only quantizing the remaining parameters.
> >
> > Specifically, they train for 1-2 epochs on large datasets (C4 for base LLaMA2 models and ShareGPT for Vicuna) to optimize both cherry and normal parameters. This additional training and the presence of full precision cherry parameters naturally lead to lower perplexities than our pure PTQ setting at the same nominal average bit-width.
> >
> > At the same time, CherryQ reports results mainly on LLama2 (7B/13B) and Vicuna-1.5, evaluated on C4 and WikiText-2. Their 2-bit experiments with average bit in 2.15-2.30 already assume a specific cherry-parameter structure (top 1/256 parameters per matrix kept at FP16) plus QAT, and obtain WikiText-2 perplexities below 10 for LLaMA2 7B/13B.
> >
> > Our work targets different model families and enforces mixed 2/4-bit without retaining FP16 cherries or applying QAT, which offers less flexibility and typically increases absolute perplexity under very low average bit budgets.
> >
> > Because of the above differences, we do not claim cross-paper comparability of absolute perplexity with CherryQ or OmniQuant. Instead, our focus is on relative performance within a controlled PTQ setting: for each model, dataset (WikiText-2), bit budget, and quantization backend (GPTQ, HQQ, Quanto), IMPQ and all baselines share:
> > 1. the same calibration data and evaluation process,
> > 2. the same memory budget derived from the target average bit,
> > 3. the same PTQ default configurations,
> >
> > The only difference across IMPQ and baselines is the layer selection strategy for mixed-precision PTQ.
> >
> > Regarding the $\textbf{W3}$, The reviewer suggests that “improving only the sensitivity is of limited benefit” and that our gains may not be credible. We emphasize that IMPQ is not purely sensitivity-based: SPQE estimates both first-order sensitivities and pairwise interaction terms at the layer level, and IMPQ solves an interaction-aware optimization problem to assign bits (Sec 3). This goes beyond simple sensitivity metrics such as weight norms or gradient magnitudes.
> >
> > Specifically, our ablations have shown that:
> > Interaction-aware assignments consistently outperform pure sensitivity-based baselines (Activation, Sensitivity, LIM, Z-Score) under the same budgets.
> > The largest relative improvements occur at the most aggressive budgets (2.01–2.5 bits), while the gap narrows as the average bit-width increases. This pattern aligns with the intuition that correctly modeling inter-layer interactions matters most when precision is extremely limited, and is unlikely to be explained solely by suboptimal baseline configurations.
> >
> > $\textbf{Response to W4}$:
> >
> > We thank the reviewer for pointing this out. Due to the large number of models and bit budgets and the strict page limit, we had to use a compact font for Table 1 in the current submission. In the final version, we will adjust the table layout so that the font size is larger and the results are easier to read.

---

### Note · Program_Chairs · 2025-12-08
**Submission Desk Rejected by Program Chairs**

Desk rejected because of hallucinated reference:
Chen Zhu et al. arXiv:2307.03744, 2023.
A survey on efficient deployment of large language models.